# Metal toxin threat in wildland fires determined by geology and fire severity

Alandra Marie Lopez [1], Juan Lezama Pacheco[1] & Scott Fendorf [1] ✉

Accentuated by climate change, catastrophic wildfires are a growing, distributed global public health risk from inhalation of smoke and dust. Under-recognized, however, are the health threats arising from fire-altered toxic metals natural to soils and plants. Here, we demonstrate that high temperatures during California wildfires catalyzed widespread transformation of chromium to its carcinogenic form in soil and ash, as hexavalent chromium, particularly in areas with metal-rich geologies (e.g., serpentinite). In wildfire ash, we observed dangerous levels (327-13,100 µg kg$^{-1}$) of reactive hexavalent chromium in wind-dispersible particulates. Relatively dry post-fire weather contributed to the persistence of elevated hexavalent chromium in surficial soil layers for up to ten months post-fire. The geographic distribution of metal-rich soils and fire incidents illustrate the broad global threat of wildfire smoke- and dust-born metals to populations. Our findings provide new insights into why wildfire smoke exposure appears to be more hazardous to humans than pollution from other sources.

Global wildfire activity represents a rising distributed health risk from smoke and dust inhalation[1–6]. While fire is a fundamental ecological process on which many ecosystems are dependent for vegetation composition, structure, and function, extreme wildfires can devastate human and ecosystem health[7]. Wildfire smoke is a complex, dynamic mixture of gases and particulate matter (PM) dependent on fire fuels and severity that activate inflammatory pathways or DNA damage responses[8–12]. Respiratory exposure to fine particulate matter (diameter ≤2.5 µm; PM$_{2.5}$) is particularly problematic owing to its ability to penetrate deep in lungs[13–15]. Recent work has demonstrated that wildfire PM$_{2.5}$ is more harmful than urban sources[16] and prescribed fires[17], each of which differ in physical and chemical properties; however, the chemical composition of wildfire PM, and its compound-specific health effects, remain poorly characterized. Many wildfire pollutants in dust and smoke are classified respiratory carcinogens; commonly studied are organic aerosols and gases derived from incomplete combustion of vegetation[8]. Increased heavy metals in PM have been documented during wildfire episodes and may induce cytotoxicity, increase lung cancer risks, and greatly contribute to oxidative stress[18–29]. In particular, chromium in its hexavalent form [Cr(VI)] represents an unforeseen health concern.

Metals in soils and ash are commonly linked to structural burning within wildland-urban interfaces (WUI)[30–33], with negligible awareness of wildland landscapes (soils and ash) as an alternative and highly distributed source[30]. Coinciding with highly populated areas globally, landscapes with ultramafic and mafic geology have soil Cr concentrations ranging from 200 up to 60,000 mg kg$^{-1}$ and average 2650 mg kg$^{-1}$ [34] (Supplementary Fig. 1). Chromium is naturally abundant in its trivalent form [Cr(III)], presenting a limited hazard to human health. However, the otherwise sluggish rate of Cr(III) oxidation by molecular oxygen accelerates at temperatures above 200 °C, leading to rapid generation of toxic Cr(VI), even from mineral-bound Cr(III) in soils[35]. On the basis of heating experiments, up to 35% of soil Cr may be oxidized to Cr(VI) within fires[35]. Urban ash collected after a wildfire in southern California, USA, contained Cr(VI) concentrations up to 3200 µg kg$^{-1}$ and naturally illustrates heat-catalyzed Cr(VI) generation[36,37]. Further, an 8.6-fold increase in Cr(VI) concentrations was observed after a controlled vegetation fire relative to pre-burn concentrations in topsoil (0-3 cm depth)[38].

Fire-affected ultramafic and mafic areas occur within all continents (excluding Antarctica), including tropical and temperate climate regions across the western U.S., Brazil, Australia, South Africa, Europe, and Indonesia, and therefore may be a natural source of metals

[1]Earth System Science Department, Stanford University, Stanford, CA 94305, USA. ✉e-mail: fendorf@stanford.edu

during fire conditions (Fig. 1). The frequency of large (greater than 2023 ha, Fig. 1) and small (Supplementary Fig. 2) fire occurrences within global mafic-ultramafic regions illustrate the propensity for global exposure to Cr(VI) within wildfire dust and smoke. Following wildfires, severely burned areas are often barren and blanketed with ash and loose, rough topsoil leading to enhanced post-fire wind and water erosion[4,39–42]. The resuspension of Cr(VI)-bearing soil dust and ash presents an acute respiratory exposure risk during and after wildfires[43,44]. Depending on PM size and Cr(VI) solubility, Cr(VI) can cause localized pulmonary toxicity and translocate to other organs[45,46], increasing risks of lung, paranasal sinuses, and nasal cavity cancers[47–49]. An excess lifetime risk of 1:100,000 is associated with an airborne Cr(VI) concentration of 0.25 ng m$^{-3}$ [50].

Fire-induced Cr(VI) production in soils and ash may represent a major pathway for human exposure to a severe toxin. Here, we examine the combination of soil properties and fire severity governing Cr(VI) formation within fire-impacted landscapes and the persistence of the toxin in surficial soil layers post-fire. The recent wildfires within California's North Coast Range, namely the 2019 Kincade Fire and the Hennessey Fire, within the 2020 LNU Lightning Complex, provided us the exceptional ability to explore Cr(VI) generation in fire-impacted landscapes of differing parent geology and under differing canopy-dependent fire severity and duration of Cr(VI) within surface soils. The generalities of our findings, however, are globally transferable.

Here, we show that fire severity, geologic substrate, and ecosystem type influence landscape-scale production of Cr(VI) in particulates during recent wildfires. We find that Cr(VI) concentrations were highest in soils derived from metal-rich geologies, including (ultra) mafic rocks and their metamorphic derivatives. Further, soil particulates from severely burned areas exhibited 6.5-fold greater Cr(VI) concentrations than from unburned soil. Reactive Cr(VI) concentrations are remarkably higher in surficial soil and ash, with up to 13,100 µg kg$^{-1}$ detected in wind-dispersible particulates (<53-µm diameter) and remained a threat in surface particulates nearly a year post-fire. Our results demonstrate that Cr(VI) persists in particulates composing smoke and post-fire dust, which presents concern for respiratory exposure of local and distal communities.

## Results and discussion

We analyzed reactive Cr(VI) concentrations in burned and unburned soils from four ecological preserves (Supplementary Fig. 3) that provide variation in ecosystems (grassland, chaparral, and forest; Supplementary Fig. 4) resulting in a gradient in burn severity (a combination of fire intensity and duration) (Supplementary Table 1)[40]. Importantly, they also span contrasting geologies (felsic to ultramafic, or metamorphic equivalents) having highly variable total Cr concentrations that range from 102 to 4070 mg Cr kg$^{-1}$ soil (Supplementary Fig. 5). Both wildfires examined in this study occurred during the late fire season (August−November), when soils were driest, and within a multi-year drought across northern California.

### Impact of geology

The geology of soil parent material affected the amount of reactive Cr(VI) concentration in fire-impacted soil (Fig. 2). We assessed burned and unburned soil profiles within rhyolitic (felsic and low Cr content; Fig. 2a), mélange (intermediate Cr content; Fig. 2b), and serpentinite (ultramafic equivalent and high Cr content; Fig. 2c) derived soils (Supplementary Table 1; Supplementary Fig. 5). The amount of reactive Cr(VI) produced in fire-affected soils increased with total Cr content and was highest in serpentine soils (Fig. 2c). Average Cr(VI) concentrations generated in soils derived from mélange (Fig. 2b) were more than double the respective levels in rhyolitic soil (Fig. 2a) regardless of overlapping ranges in total Cr content, 152-954 and 102−338 mg kg$^{-1}$, respectively, reflecting the potential contribution of differing mineralogy to Cr(VI) generation (Supplementary Fig. 6)[35]. Except in near surface depths, reactive Cr(VI) concentrations were below the detection limit in rhyolitic soils (Fig. 2a).

Reactive Cr(VI) concentrations were higher at near-surface depths (0−2 cm) in burned soil across all geology types with respect to near surface unburned soil and within the soil profile at control depths

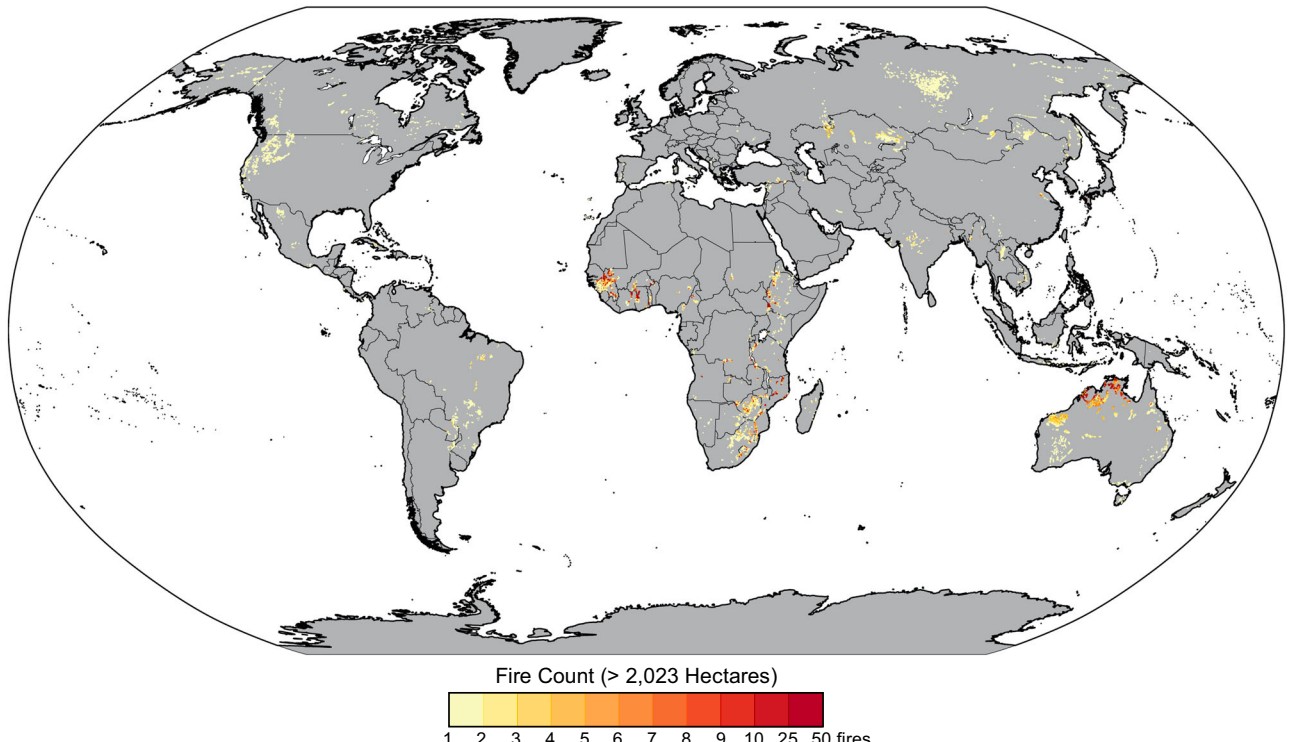

**Fig. 1 | Frequency of large fires in metal-rich landscapes.** Satellite-derived fire occurrences greater than 2023 ha within generalized mafic and ultramafic landscapes from 2001 to 2020.

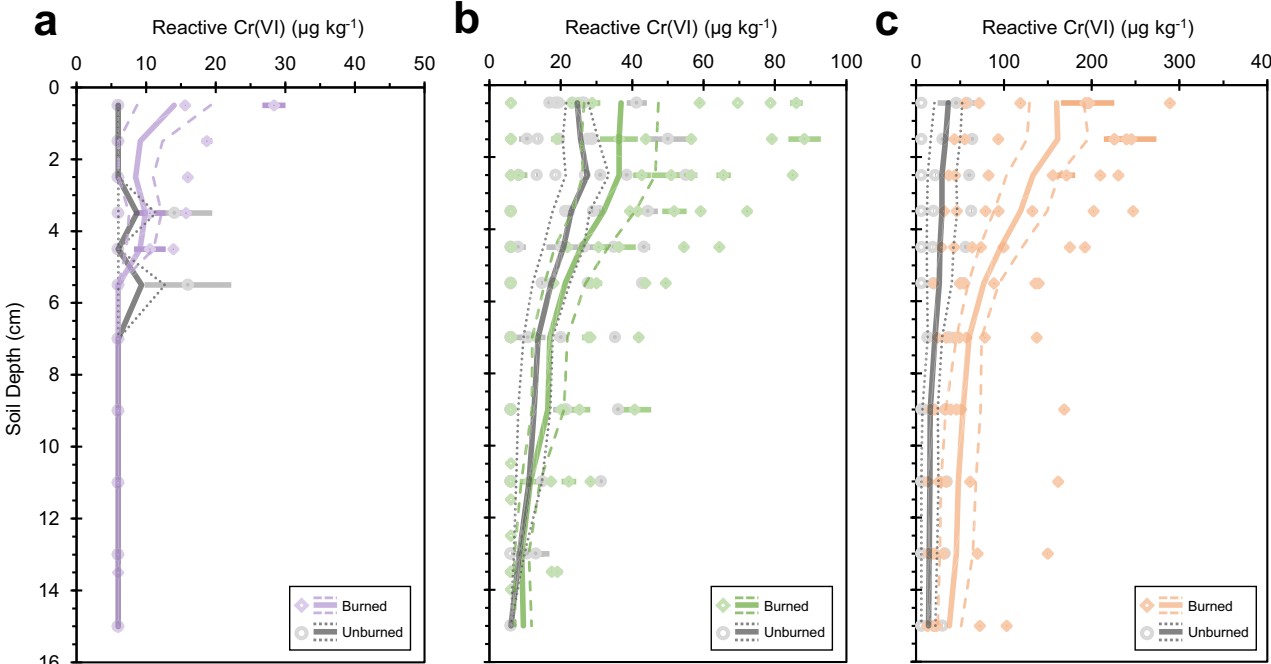

**Fig. 2 | Effect of geologic substrate on fire-catalyzed Cr(VI) formation in soil.** Average reactive Cr(VI) concentrations ($\mu g\,kg^{-1}$) within (**a**). rhyolite-, (**b**). mélange-, and (**c**). serpentinite-derived soil profiles (0−16 cm) that were not burned (gray circles; rhyolite, $n = 3$; mélange, $n = 7$; serpentinite, $n = 3$) or fire-affected (colored diamonds; rhyolite, $n = 4$, purple; mélange, $n = 9$, green; serpentinite, $n = 7$, orange). Note that the x-axis bounds differ by panel. Points are reactive Cr(VI) concentrations for each soil core based on triplicate measurements (error bars represent standard error). Concentrations below the detection limit are plotted as half the detection limit ($6\,\mu g\,kg^{-1}$). Solid and dashed lines represent the mean (based on $n$ soil cores) and standard error of the means, respectively.

(10−16 cm). Depending on burn severity, combustion and heat transfer drive the development of steep thermal gradients, with the highest soil temperatures occurring in the top 5-cm, below which temperatures rarely exceed 150 °C[40]. Despite a range of fire severity conditions (Fig. 3), serpentine soils had the greatest difference in reactive Cr(VI) concentrations at near surface depths in burned versus unburned soils (p-value = 0.041, Supplementary Table 2). Reactive Cr(VI) concentrations at control depths did not significantly differ between burned and unburned serpentine soils (p-value = 0.25, Supplementary Table 3) and neither concentration within unburned soil profiles (p-value = 0.37, Supplementary Table 4). In fact, Cr(VI) concentrations showed little variability between burned and unburned sites and between geologic conditions at deeper control depths despite differences in total Cr content (Supplementary Tables 2 and 3). Moreover, rhyolitic and mélange sites experienced low severity fire conditions during the Kincade Fire and the concentration differences between near surface burned and unburned soils were not statistically significant (Supplementary Table 2).

### Impact of fire severity

We examined a gradient in fire severity conditions within a Cr-rich serpentine chaparral landscape (Fig. 3). Average Cr(VI) concentrations in severely burned sites exceeded $200\,\mu g\,kg^{-1}$ in near surface soil (0−2 cm), which was 6.5-fold greater than Cr(VI) concentrations in soil of unburned chaparral landscapes ($35\,\mu g\,kg^{-1}$, p-value = 0.0001, Supplementary Table 2) and 3.6 times more than at control soil depths below 10 cm within burned soil cores ($63\,\mu g\,kg^{-1}$, p-value = 0.0109) (Fig. 3a, Supplementary Table 5). In low severity soils, the average reactive Cr(VI) concentration was slightly elevated compared to soil in unburned areas, 73 and $35\,\mu g\,kg^{-1}$ within the top 2-cm (Fig. 3a), respectively, though this difference was not statistically significant (p-value = 0.173, Supplementary Table 2). Similarly, Cr(VI) concentrations were not statistically different across depths within low severity soil cores (Supplementary Table 5). Average reactive Cr(VI) levels

within unburned soil cores ranged from non-detectable to $64\,\mu g\,kg^{-1}$ and were consistent with water-soluble concentration profiles in near surface soil from a proximal serpentine grassland sampled in 2014 with up to $27\,\mu g$ Cr(VI) $kg^{-1}$[51].

Independent of geology, we observed low fire severity results in minimal Cr(VI) generation, and the reactive fraction was not significantly different from unburned soil profiles (Figs. 2, 3a). Our field observations suggest that low severity fires do not generate Cr(VI) concentrations of concern even in soils with high total Cr concentrations (Supplementary Table 1). At the contrasting end of the fire temperature range, previous work on soil[35] and industrial ash has documented that mineral and Cr transformations above 800 °C can also limit (reactive) Cr(VI) production[52]. In a Knobcone pine (*Pinus attenuata*) forest that experienced extreme soil heating (>800 °C), we observed lower reactive Cr(VI) concentrations in near surface soil (Supplementary Fig. 7). The effects of fire severity on chemical differences highlight the temperature dependence of soil Cr(VI) production.

### Contribution of ash

Blanketing the ground in the burned serpentine chaparral was a surface layer that contained disaggregated ash and soil minerals highly susceptible to wind erosion compared to the underlying, structured soil[53]. Rhyolitic and mélange sites did not have distinguishable surficial soil-ash layers because of the lower severity fire conditions. Total Cr content varied from 999 to 4829 mg $kg^{-1}$ (Supplementary Table 6). We observed reactive Cr(VI) concentrations as high as $3335\,\mu g\,kg^{-1}$ in the bulk composition (particles less than 2-mm diameter) of the surficial soil-ash layers (Fig. 3b). At moderate-high fire severity sites, surficial soil-ash layers contained an average reactive Cr(VI) concentration of $1073\,\mu g\,kg^{-1}$, which was nearly 5-fold greater than the average concentration in the top 1-cm of soil ($219\,\mu g\,kg^{-1}$; Fig. 3a). These Cr(VI) concentrations exceeded the United States Environmental Protection Agency (US EPA) recommended screening level for residential soils,

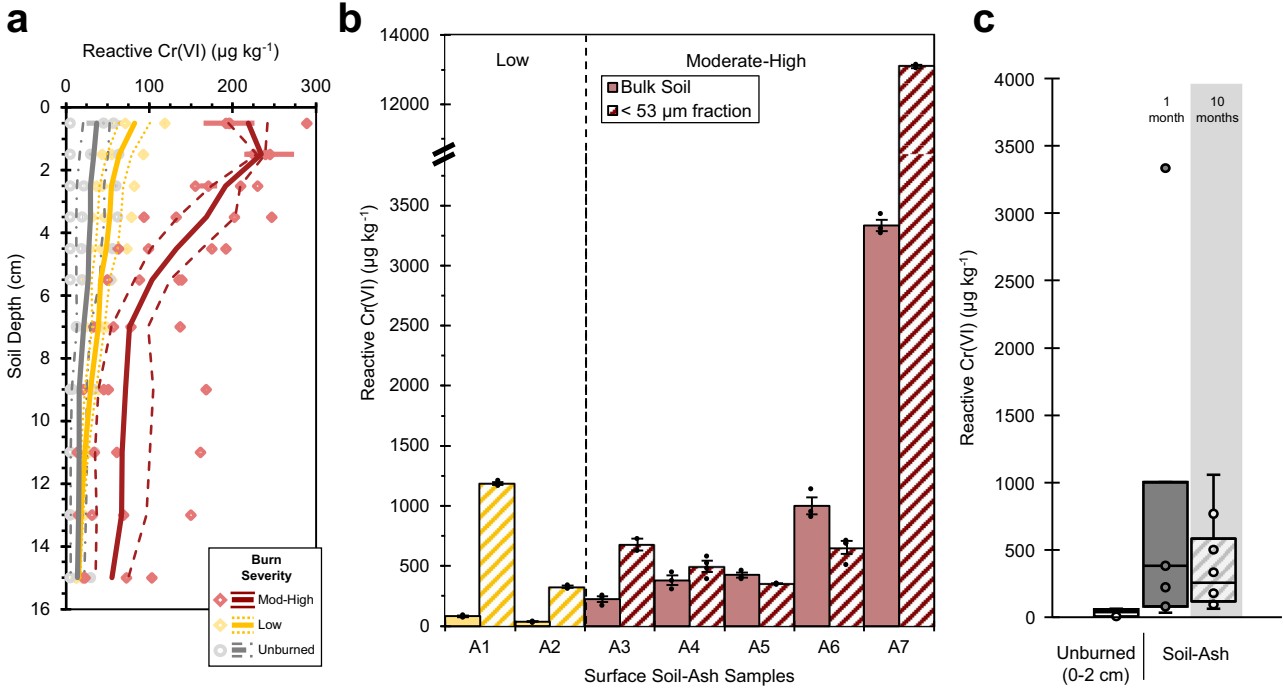

**Fig. 3 | Effect of fire severity on Cr(VI) formation and persistence in soil and ash.** **a** Average reactive Cr(VI) concentrations (µg kg⁻¹) within soil profiles (0–16 cm) that were not burned (gray, $n = 3$), experienced low (yellow; $n = 3$) or moderate-high burn severity (red; $n = 4$) in a serpentine chaparral landscape. Points (gray circles, unburned; yellow diamonds, low severity; red diamonds, moderate-high severity) are reactive Cr(VI) concentrations for each soil core based on triplicate measurements (error bars represent standard error). Solid and dashed lines respectively represent the mean (based on $n$ soil cores) and standard error of the means. **b** Average reactive Cr(VI) concentrations (µg kg⁻¹) from bulk surface soil and ash layers (<2 mm; solid) and the sieved fraction of silt and clay-sized particles (<53 µm; striped) collected in low (yellow) and moderate-high (red) burn severity locations. Error bars represent standard error of the mean and points are individual replicates. The y-axis is hatched to include reactive Cr(VI) concentration measured in the <53-µm size fraction of A7. **c** Box and whisker plot of reactive Cr(VI) concentrations (µg kg⁻¹) within the surface (0–2 cm) of unburned serpentine soil ($n = 6$) and from bulk surface ash layers collected in low and moderate-high burn severity locations 1-month post-fire (September 2020, $n = 7$) and 10 months post-fire (July 2021, $n = 10$). The center line is the median, the box spans the interquartile range, the whiskers are 1.5x interquartile range, and the points represent individual samples, including outliers.

300 µg Cr(VI) kg⁻¹, based on its carcinogenicity[54]. At site A7, the average reactive Cr(VI) concentration was more than three times greater than other moderate-high fire severity sites (Fig. 3b). We suspect that longer burning duration and fire intensities with greater biomass combustion contributed to the relatively high levels of Cr(VI), as this was a severely burned forested area. Importantly, ash from severely burned areas concentrate alkali (Na, K) and alkaline earth (Ca, Mg) metals (often from biomass combustion) that are key for the thermal oxidation of Cr(III)[9,52]. For example, CaCrO₄ was noted after agricultural soil amended with composted Cr(III)-rich tannery sludge was heated to 500 °C[55].

At low burn severity sites, average reactive Cr(VI) concentrations in the surficial soil-ash layer were 36 and 82 µg kg⁻¹ (Fig. 3b), approximately 3–40 times less than bulk measurements collected from moderate-high severity sites that ranged from 222 to 3335 µg Cr(VI) kg⁻¹. The physicochemical conditions within the soil-ash layer also contribute to the longevity of Cr(VI)[56]. Due to incomplete biomass combustion, ash from low severity fires is characteristically rich in pyrogenic carbon, which increases Cr(VI) reducing capacity and promotes Cr immobilization as mineral-bound Cr(III)[9,57,58]. Reactive Cr(VI) concentrations in low severity surficial soil and ash were consistent with levels present in the near-surface depths of low severity and unburned soil (Fig. 3).

### Persistence of chromium(VI)
With minimal rainfall, reactive Cr(VI) persisted within the surface soil-ash layer ten months after post-fire sampling (July 2021) (Fig. 3c). At McLaughlin Natural Reserve, precipitation was merely 25.1 cm (September 2020–July 2021) since the Hennessey Fire, which is

approximately a third of the average precipitation (74.3 cm) during these months[59]. After nearly a year, the serpentine chaparral remained covered with loose surface soil and ash (approximately 2–5 cm thick), illustrating the absence of significant water erosion or leaching of reactive Cr(VI). Reactive Cr(VI) concentrations ranged from 64 to 1060 µg kg⁻¹, with a median concentration of 257 µg kg⁻¹, and remained elevated compared to concentrations (5–64 µg kg⁻¹) within the near surface depths (0–2 cm) of unburned serpentine soil (Fig. 3c); however, these differences were not statistically significant ($p$-value = 0.164).

### Wind-dispersible fraction
Within severely burned landscapes, wind velocities and dust flux are enhanced post-fire, and conditions may persist for multiple years during droughts and depending on revegetation rates[4,44,53]. Wind-driven suspension of dust is a major mechanism of soil-borne contaminant transport in semiarid environments after wildfires[5,43,44]. Similar to soil nutrients and other contaminants, the reactive fraction of Cr(VI) is often associated with fine, wind-dispersible particles, primarily clay-sized particles less than 2-µm diameter, due to their high surface area that dominates adsorption sites[5]. Moreover, post-fire soil pH may exceed 8.5, depending on ash content and fire severity, which increases Cr(VI) stability and enhances extended Cr(III) oxidation by O₂ under ambient temperature conditions[56]. Burning of a Pinus forest on ultramafic terrain illustrates the susceptibility of fine particles to wind transport post-wildfire, where surficial soil and ash high in Cr (and Ni) were lost to wind erosion[60].

Surface soil-ash layers across sites contained 12–22% particles smaller than 53-µm diameter, representing wind-dispersible silt- and

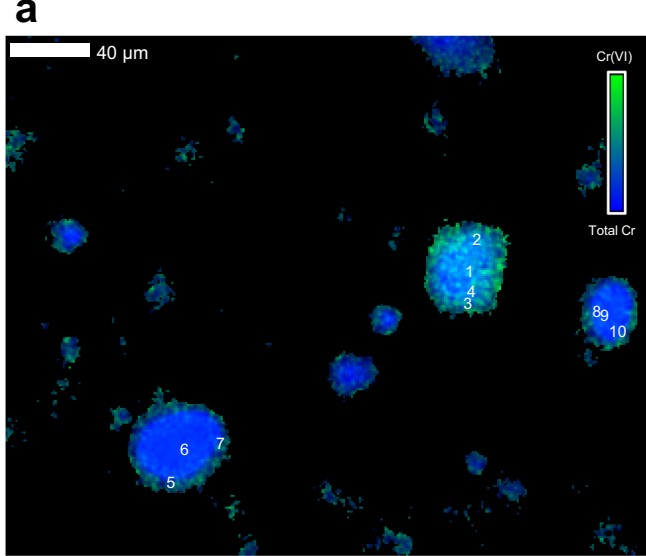

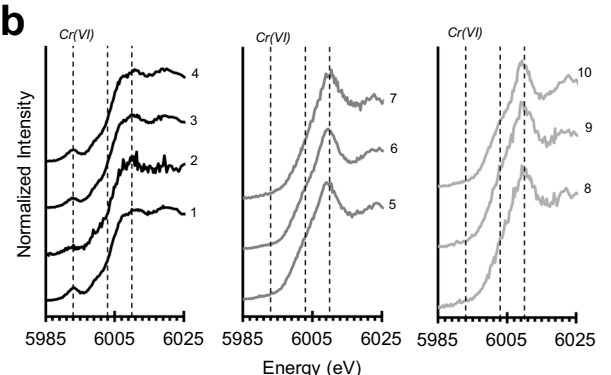

**Fig. 4 | Total Cr(VI) in wind-dispersible soil and ash particles. a** μ-XRF image (pixel resolution: 1 μm) showing the relative intensity of Cr(VI) (green; estimated as the intensity ratio at 5993 and 6010 eV) and total Cr (blue; measured at 6010 eV) within the <53-μm size fraction of Cr-bearing soil-ash particulates from a serpentine chaparral that experienced high fire severity (A7). **b** Normalized μ-XANES spectra (Cr K-edge) from numbered locations on Cr-bearing particles in a. Dashed lines indicate energies characteristic of Cr(VI) (5993 eV), Cr(III) (6003 eV), and total Cr (6010 eV), at which μ-XRF images were also collected.

clay-sized particles (Supplementary Table 6). Respirable particles less than 10-μm diameter made up approximately 27–30%, of which 8–12% were less than 2.5-μm diameter. Total Cr content (946–1643 mg kg⁻¹) was less than half the total Cr concentration measured in respective bulk compositions (up to 2-mm diameter). Chromium has been detected in nanoparticles of structural ash ranging from 90 to 270 nm in size[61]. Concentration ranges of additional metals enriched in ultramafic rocks, Ni, Fe, and Mn, were 2294–3489 mg kg⁻¹, 82.7–105 mg g⁻¹, and 1910–2547 mg kg⁻¹, respectively. A recent study found greater total Fe content in wildland ash (with variable geologic substrate) than structural ash, with median concentrations of 36.1 mg g⁻¹ and 18.5 mg g⁻¹, respectively[61].

Using micro-scale X-ray techniques, Cr(VI)-containing soil and ash particulates were identified in a high fire severity sample but not in a low fire severity sample (Fig. 4; Supplementary Fig. 8). Here, Cr(VI) was associated with mineral surfaces (e.g., adsorbed) or enriched in relatively low-Cr particles with Ca and K (Fig. 4, Supplementary Fig. 9). Consistent with particle analysis, reactive Cr(VI) concentrations spanned from 327 to 13,100 μg kg⁻¹ (Fig. 3b). Reactive Cr(VI) concentrations were similar or higher than respective concentrations in bulk soil and ash (up to 2-mm particle size) from moderate to high fire severity sites

in the serpentine chaparral, and all of which exceeded the US EPA screening level (300 μg kg⁻¹). Interestingly, we detected 9 to 14 times higher reactive Cr(VI) concentrations in the inhalable size fractions of surface soil and ash from low fire severity sites (Fig. 3b).

Suburban fires illustrates the impacts of inhaling Cr(VI)-containing ash within the respiratory tract by measuring Cr(VI) leached with a simulated lung fluid[36,37] and discerning Cr mineralogy within nanosized particulates (<100 nm)[27,29]. Our results indicate that comparable toxic levels of reactive (water-soluble) Cr(VI) are formed within a serpentine wildland landscape as previously observed from structural burning[36,37] (Fig. 3b), and stresses that Cr(VI) is enriched in wind-dispersible particles of surface soil and ash. Exposure to particulate Cr(VI) is thus a concern both during and subsequent to high-severity wildland fires on mafic/ultramafic substrates. In addition to local (and possibly distal) communities that may be exposed by transport of Cr(VI)-bearing PM during and after wildfires, wildland fire crews and first responders are particularly at risk. Given the persistence of Cr(VI) after wildfires, continued dust exposure serves as a continued human health threat to land workers, recreationalists, and local residents.

While recognized for urban fires, threats from metal exposure in smoke and dust need to be recognized within wildland fires arising on metal-rich geologies. Across tropical climate regions, deeply weathered lateritic soils are common, in which Cr is predominantly found within crystalline Fe oxides (e.g., hematite, goethite), and soil Fe content may exceed 500 mg g⁻¹ [62]. Past work has quantified the Cr oxidation capacity of Cr-bearing Fe oxides and lateritic soils during heating simulations[35,62]. For example, the greatest Cr(VI) formation (more than 40% of total Cr) upon heating hematite occurred at temperatures less than 400 °C, while up to 100% of total Cr in goethite transformed to Cr(VI) at 800 °C. In Mediterranean or temperate climates (similar to this study's region), ultramafic soils are relatively more enriched in Cr-bearing phyllosilicate minerals (e.g., serpentine), Fe oxide content is moderate (typically less than 100 mg g⁻¹) with more amorphous secondary phases, and soil pH is neutral to alkaline pH. In our study, and under natural wildfire conditions, we observed that up to about 0.015% of total Cr was reactive Cr(VI) in burned serpentine soils (Supplementary Fig. 6). Chromium(VI) generation during wildfires depend on fire conditions and host mineralogy; thus, the extent of Cr(VI) formation in lateritic soils may differ from temperate serpentine soils, but field observations of Cr(VI) in burned lateritic soils following wildfires are currently lacking.

Similarly, the generation of Cr(VI) within PM is particularly concerning and is long-lived within surficial soil-ash layers, leading to a persistent exposure risk during and after fires. With wildland fires expected to increase in frequency and severity in many geographic regions due to a combination of climate change and past fire management[63,64], post-fire dust emissions are likely to rise[41]. Thus, recognizing chemical-specific health risks beyond general PM₂.₅ is crucial. Information about environmental drivers of fire-catalyzed Cr(VI) generation in wildland areas can help guide the development of prediction tools to mitigate exposure risks to first responders and local communities. Further, the threat is global, with all vegetated continents being threatened by the combined impacts of increasingly severe wildfires across metal-rich landscapes.

## Methods

### Study area

Soils and ash were collected from the North Coast Range, which has a Mediterranean climate, across Sonoma, Napa, and Lake Counties, California, U.S.A. after the Kincade (38.7925°, −122.7801°) and Hennessey Fires (within the LNU Lightning Complex; 38.4819°, −122.1486°) in November 2019 and September 2020, respectively. Our field sites were located at four natural preserves that were partly or completely burned during one of the wildfires (Supplementary Fig. 2): Pepperwood Preserve (1295 hectares; 38.5698°, −122.6917°), Sonoma Land

Trust's White Rock Preserve (37 hectares; 38.7082°, −122.8332°), Audubon Canyon Ranch's Modini Preserve (1214 hectares; 38.7069°, −122.7625°), and University of California's Donald and Sylvia McLaughlin Natural Reserve (2833 hectares; 38.8712°, −122.4215°).

Pepperwood, White Rock, and Modini Preserves are located within the 31,468-hectare burn scar of the Kincade Fire. The McLaughlin Natural Reserve burned during the Hennessey Fire (123,692 hectares) (Supplementary Fig. 2). Burn severity at each sampling site was determined based on landscape fire history, post-fire observations of aboveground vegetation, ash and soil color, and was further supported by laboratory measurements, including bulk soil mineralogy and ammonium ($NH_4^+$) concentrations (Supplementary Fig. 10), a direct combustion product[40]. Pepperwood Preserve, Modini Preserve, and McLaughlin Natural Reserve had partly or completely experienced at least one wildfire since 1984[65]. Pepperwood Preserve completely burned (95%) during the 2017 Tubbs Fire, which was a near identical footprint to the preceding Hanly Fire in 1964. Modini Preserve was previously burned during the 2004 Geysers Fire, and McLaughlin Natural Reserve previously burned in 1999 and 2015.

Vegetation classifications at each field site were determined using countywide map databases (Supplementary Fig. 3, Supplementary Table 1)[66,67]. At McLaughlin Natural Reserve, the sampled serpentine chaparral was dominated by post-fire seed-regenerating species, including whiteleaf manzanitas (*Arctostaphylos viscida*) and Macnab's cypress (*Hesperocyparis macnabiana*), and post-fire resprouting species, including leather oak (*Quercas durata*) and musk brush (*Ceanothus jepsonii*). The chaparral experienced two recent, major wildfires in 2015 and 2020, which resulted in overlapping mosaic burn patterns. Low severity sampling sites had burned twice in the five-year period and contained a limited shrub cover in 2020. Surface temperatures were estimated to not exceed 250 °C[40]. Sites that were classified as moderate to high burn severity consisted of dense whiteleaf manzanita stands that did not burn in 2015, resulting in an accumulation of highly volatile fuels. Within the moderate-high severity areas, bare, disaggregated reddened soil crosscut a characteristic thick layer of black and white ash[39]. Soil temperatures at the surface likely exceeded 700 °C as reported in a previous study on soil heating in chaparral ecosystems during various fire severities[68]. Within the surface soil-ash layer, forsterite was detected (Supplementary Fig. 11), indicating that surface temperatures exceeded 575−600 °C for sufficient time to thermally decompose serpentine minerals (e.g., lizardite) present within surface soil[69,70].

Sampling sites at Modini Preserve were located within grasslands and a densely vegetated forest. The grasslands are perennial with native purple needle grass (*Stipa pulchra*) and blue wildrye (*Elymus glaucus*) and invaded by annual non-native species. The forest contained fire obligate knobcone pines (*Pinus attenuata*) with an understory dominated by whiteleaf manzanitas, chamise (*Adenostoma fasciculatum*) and resprouting madrone trees (*Arbutus menziesii*). Like Modini, two field sites at Pepperwood Preserve were within grasslands that are primarily annual exotics (*Bromus hordeaceus, Avena barbata, Festuca perennis, Elymus caput-medusae*, and *Festuca bromoides*) with an abundant diversity of native perennial grasses. Since 2013, cattle grazing (up to 250−300 heads annually since 2018) has played a major role in reducing fuel loads. While grassland fire intensities can be high (depending on their moisture content), fires often burn rapidly (short duration) limiting overall burn severity and the exposure of soils to high temperatures[39]. Fire progression and its effects on grasslands is also often uniform, unlike the mosaiced burn pattern observed in chaparral. Grassland sites examined in this study had low severity fire as exhibited by post-fire charred litter and grass stems that were not completely combusted. Additional field sites at Pepperwood Preserve were located along the boundaries of a mixed hardwood forest and a savanna of blue oak (*Quercus douglasii*) underlain by grasses, and within a forest of Douglas fir (*Pseudotsuga menziesii*). At White Rock

Preserve, soil sampling sites were located within grasslands and in a valley oak (*Quercus lobata*) forest underlain by grasses.

The four natural preserves in this study comprise contrasting geologies, including felsic (rhyolite, rhyodacite), mélange (consisting of mixed geology, including greenstone and metabasaltic blueschist, along with graywacke), and ultramafic (serpentinite) rocks, which are characteristic of the North Coast Range (Supplementary Fig. 4, Supplementary Table 1). The geology at each sampling location was determined based on surface bedrock characterization using detailed regional geologic maps when available[71–75]. Regional sandstone (*Kfss*) comprises the Devils Den Terrane of the Franciscan Complex and contains a multitude of rocks, including graywacke, derived from oceanic crust of the Late Cretaceous[71]. Further, serpentinite results from the Coast Range Ophiolite, which reaches 320 km across the Coast Range. Sandstone and serpentine soils contained 686−1518 and 1137−4070 mg Cr kg$^{-1}$, respectively (Supplementary Table 1). Mélange (*fsr, Kjfs*) of the Franciscan Complex consists of a complex mixture of rocks due to tectonic activity, and associated soils (152−954 mg Cr kg$^{-1}$) were located at White Rock and Modini Preserves. At Pepperwood Preserve, fire-affected soils with low Cr content (102−338 mg Cr kg$^{-1}$) were collected from areas with silica-rich rhyolitic and rhyodacitic rocks (*Tsr, Tst*, and proximal surface deposits).

Major soil types are described in the U.S. Department of Agriculture Natural Resources Conservation Service's Soil Survey[76]. Soil sampling sites within the serpentine chaparral at McLaughlin Natural Reserve were clay loam to very gravelly clay loam within the top 30-cm. At Pepperwood Preserve, rhyolitic soils were very gravelly loam and clay loam. Mélange-derived soils at White Rock and Modini Preserves were clay loam, and sandstone soils at Modini Preserve were gravelly loam to clay loam.

## Soil and ash sample collection
The three preserves (Modini, White Rock, and Pepperwood Preserve) that burned during the Kincade Fire were sampled twice, once immediately following the fire's complete containment (October 23−November 6, 2019) on November 15th and 19th, 2019, and then approximately 11 months post-fire on September 18th, 2020. Sampling sites at McLaughlin Natural Reserve were sampled once on September 19th, 2020, before full containment of the LNU Lightning Complex (August 17−October 2, 2020), and bulk surface soil and ash were resampled on July 29th, 2021. Immediate post-fire soil and ash sampling was completed before any major precipitation events to minimize Cr(VI) transport and prevent blurring of fire effects on the physicochemical composition within the top 30-cm of soil profiles.

Near surface soil samples were collected from fire-affected and unburned locations as intact cores using a slide hammer soil core sampler (5 × 30 cm; AMS Inc., ID, USA) and cased in plastic liners with caps. Soil cores ranged from 15 to 30 cm and depended on the refusal depth at each sampling location. A total of 38 cores were collected across the four preserves, 22 from burned areas and 16 from unburned locations. At McLaughlin Natural Reserve, unburned soil cores were collected from locations that hadn't been burned in at least 5 years. Unburned controls at Pepperwood, White Rock, and Modini Preserves were collected approximately 11 months post-fire from sampling sites that burned at low severity during the Kincade Fire. Among all sampling locations, 19 cores were collected from grasslands, 10 cores were from chaparral, and 9 cores from mixed forests. Ten cores were from serpentine (ultramafic) soil, 7 from rhyolitic (felsic) soil, 16 soil cores were from mélange (similar chemistry as mafic) parent material, and 5 were from sandstone (similar chemistry as mafic). Within the severely burned serpentine chaparral at McLaughlin Natural Reserve, bulk soil and ash were collected from the loose, surface soil overlying the coring. Depending on fire severity, this layer contained charred plant material, black to white ash, and disaggregated reddened soil. Prior

to laboratory analysis, field moisture was retained by storing cores and bulk samples within air-tight containers at 4 °C.

## Sample preparation and solid-phase analysis

The top 6-cm of the soil cores were manually divided into 1-cm intervals and the remaining depths were sliced in 2-cm intervals to an analysis depth of 16-cm. Except for extremely severe burned soils (Supplementary Fig. 7), we considered soil depths below 10 cm as a control depth[36,38]. Each soil section and bulk sample from surface soil-ash layers was homogenized, and soil aggregates were lightly ground in an agate mortar and pestle to pass through a 2-mm sieve. Gravimetric water content was determined by the weight difference of field-moist and oven-dried soil (105 °C for 24 h). Within the top 16-cm of the soil, the gravimetric water content ranged from 1 to 11%.

We passed representative samples of bulk surface soil-ash layers through a series of sieves to separate ash/soil solids into two size fractions: sand-sized (0.053−2 mm) and silt/clay-sized particles (less than 53 μm). The additional sieving to collect particles less than 53 μm allowed us to further quantify the percentage of this inhalable size fraction and analyze its physicochemical characteristics. We did not gently grind soil prior to sieving so as to preserve sand-sized micro-aggregates that would otherwise result in an overestimation of the fine particle fraction highly susceptible to dust resuspension. Particle-size distribution (0.4−53 μm) of the silt/clay-sized particle fraction was determined using a laser diffraction particle size counter (Coulter LS-230) for select moderate and high severity field site samples (A4, A6, A7). Homogenized, air-dried samples were prepared in a suspension with $2\,g\,L^{-1}$ sodium hexametaphosphate solution and vortexed for 25 min. Particle-size distributions were determined by averaging two measurements for each sample.

Total major and trace element concentrations of finely ground oven-dried soil samples were quantified by energy dispersive X-ray fluorescence (XRF) spectrometry (Spectro XEPOS). To assess thermal alterations in bulk mineralogy within the surface soil ash layers and soil profiles from burned sites, a subset of soil aliquots was dried at room temperature and qualitatively analyzed by powder X-ray diffraction (XRD; Rigaku MiniFlex 600) with Cu-Kα radiation (40 kV, 15 mA). Diffraction patterns were collected in the 2θ range of 3−90° with a 0.01° step size, 1.2 s time count per step. The XRD patterns were evaluated using PDXL 2 software (Rigaku).

## Micro X-Ray fluorescence analysis

Fine particulates (<53 μm) from two surface soil-ash layers (representing high and low fire severity) were fixed in epoxy resin (EPOTEX 301-2FL) and thin sections (30-μm thickness) were prepared on quartz slides (Grindstone Laboratory LLC, Portland, OR). Thin sections were analyzed in ambient air conditions and in fluorescence mode at the bending magnet beam line 2−3 at Stanford Synchrotron Radiation Lightsource (SSRL, Stanford, CA) with a double Si(111) crystal monochromator to tune the X-ray beam and a Si Drift Detector (Vortex-90EX, Hitachi) for data collection. The size of the X-ray beam was focused to 1-μm diameter (Sigray). Data processing and visualization was completed using the Microanalysis Toolkit software package[77].

Micro-XRF maps were collected with a 10-μm (diameter) spot size across approximately 2−4 mm grids of each thin section at multiple incident energies to examine the micron-scale distribution of Cr, Fe, Mn, and other light elements (e.g., Ca, K). Chromium speciation maps were generated at 5993, 6003, and 6010 eV, energies characteristic of Cr(VI), Cr(III), and total Cr, respectively. The X-ray beam was calibrated by measuring the X-ray absorption near-edge structure (XANES) spectral region of a Cr metal foil, and by setting the position of the first maximum of its first derivative to 5989 eV. The pre-edge absorption peak of $K_2CrO_4$ was set to 5993 eV. We also collected a coarse-scale map above the Mn and Fe K-edges at 7250 eV. We selected regions within the coarse-scale maps to collect finer resolution μ-XRF maps

$(1 \times 1\,\mu m$ spot size) at 5993, 6003, and 6010 eV. These regions were chosen based on principal component analysis and correlations with other elements.

The relative abundance of Cr(VI) in particulates of surface soil-ash layers was approximated within mapped regions. For each image, we removed pixels along edges and dead pixels, and applied a smoothing function ("blur") to each energy channel, which takes a Gaussian distribution of pixel values within a 5 × 5 pixel area (standard deviation of 0.8) to minimize background noise. Using the map generated at 6010 eV, we defined Cr-bearing particulates using an inverse-binary threshold of approximately 1% (0.56−1.69%), which assigns any pixels with intensities less than 1% of the highest intensity as background in the mapped region. Thereafter, we calculated the ratio of fluorescence intensities at 5993 and 6010 eV within identified particulates to estimate the presence of Cr(VI). Previous work has shown that the height of the Cr(VI) pre-edge peak is proportional to the total concentration of Cr(VI)[78]. The presence of Cr(VI) yields a strong absorption peak at 5993 eV; however, distortion of the octahedral Cr(III) environment can result in two weaker peaks at 5990 and 5993 eV, with the peak intensity being relatively larger at 5990 eV.

To corroborate bulk measurements of reactive Cr(VI), we combined multi-energy mapping with μ-XANES of select spots on particles to confirm the presence of Cr(VI). We acquired Cr μ-XANES spectra for points across the high-resolution particle maps, which were collected from −230 to 270 eV around the Cr K-edge. Up to 9 spectral scans were collected to increase the signal-to-noise ratio for particles with relatively low Cr concentrations. XANES spectra were processed in the Microanalysis Toolkit and ATHENA software packages.

Although μ-XRF imaging provides a unique ability to spatially speciate Cr-bearing particles, it is not without limitations. First, the size fraction of greatest concern for respiratory exposure are particulates less than 2.5 μm, which is near or below the pixel resolution for μ-XRF imaging. By setting an inverse-binary threshold to isolate Cr-bearing particulates, it's likely that we removed inhalable Cr-bearing particles (<10 μm) of relatively low intensities. Second, we underestimate Cr(VI) for particles highly enriched in Cr, having an overwhelming abundance of Cr(III).

## Aqueous extractions and chemical analysis

Reactive Cr(VI) concentrations (most available fraction, including dissolved and adsorbed Cr(VI)) in soil-ash samples (bulk and particle size fraction <53 μm) and within soil cores were extracted with 10 mM $K_2HPO_4/KH_2PO_4$ (buffered at pH 7.2)[79,80]. Phosphate effectively competes with Cr(VI) ions for surface adsorption sites. At circumneutral to alkaline pH ranges in natural soils, it's expected that nearly all aqueous Cr is present in the hexavalent form, and that Cr(VI) concentrations will be primarily limited by adsorption[81]. The clay size fraction (less than 2-μm diameter) typically has a dominant influence on species retention given their high surface areas and greater number of adsorption sites; therefore, it is likely that the reactive Cr(VI) concentrations measured here largely represent the fraction of Cr(VI) associated with clay particles. Triplicate samples were agitated in a 1:4 soil/solution ratio for 24 h, centrifuged (30 min, 2272 g, 4 °C), and filtered through 0.22-μm filters. A subsample of unacidified filtrate was used to quantify aqueous Cr(VI) concentrations using the diphenylcarbazide (DPC) method on a UV-Vis spectrophotometer (Shimadzu UV-1601)[79,82]. The detection limit was $3\,\mu g\,L^{-1}$ (approximately $12\,\mu g\,kg^{-1}$)[82]. Total Cr concentrations were determined with inductively coupled plasma mass spectrometry (ICP-MS, Thermo Scientific XSERIES 2), and confirmed that approximately all aqueous Cr was in the form of Cr(VI) in unburned soil and burned soil and ash, similarly observed in previous studies[36,37]. An aliquot of each soil extract was immediately acidified post-filtration and stored in 2% nitric acid at 4 °C until ICP-MS analysis.

To determine relative differences in $K^+$-extractable $NH_4^+$ concentrations (mg $NH_4^+$-N $kg^{-1}$) within burned and unburned soils

(Supplementary Fig. 10), additional unacidified samples (after $K_2HPO_4$/ $KH_2PO_4$ extraction) from 30 of the 38 total soil cores (21 fire-affected and 9 unburned soil cores) were frozen at −20 °C until chemical analysis. Ammonium is a direct combustion product and will be elevated in the near surface soil after wildfires depending on burn severity[40]. Ammonium concentrations in the top 6-cm were measured in triplicate (when sample volume allowed) using a flow injection analyzer (Westco SmartChem 200 Discrete Analyzer), with a detection limit of 0.05 mg $L^{-1}$ [82].

## Statistical analyses

Means and standard errors were calculated for aqueous and solid-phase chemical measurements in all cores using replicates described below. Half the detection limit was used when measured concentrations were below detection limits. Total elemental concentrations were measured in 3–4 solid-phase aliquots from each soil core (Supplementary Table 1). At each soil depth interval (1-cm from 0 to 6 cm; 2-cm from 6 to 16 cm), triplicate aqueous extractions were conducted to evaluate reactive Cr(VI) and exchangeable $NH_4^+$ concentrations (Fig. 2, Fig. 3a, Supplementary Fig. 10). In select soil depths within cores, replicates were limited (less than 3) due to solid mass or post-extraction aqueous volume.

To assess data normality, we applied the Shapiro-Wilk test and reported $W$ statistics and p-values (Supplementary Table 7). If data met normality assumptions at the 95% confidence interval ($p$-value = 0.05), we used two-sided parametric tests; otherwise, we utilized two-sided nonparametric tests. Likewise, we used the f-test to determine equal variance. Unpaired $t$ tests were used to compare mean reactive Cr(VI) concentrations at the 95% confidence interval in near surface soil (0–2 cm) of fire-affected and unburned sites based on geology. If one or both datasets were not normally distributed, such as in burned and unburned soils at control depths (10−16 cm), Mann−Whitney $U$ test was used. Within a soil core, we compared mean reactive Cr(VI) concentrations in surface soil (0–2 cm) versus control depths (10−16 cm) using either paired $t$ test or Wilcoxon signed rank test. Detailed information about and results for each statistical analysis is provided in Supplementary Tables 2−5 and 7. All statistical analyses were executed using the *stats* package in R (v. 4.1.3).

## Global fire frequency map

Wildfires burn millions of wildland hectares each year. To approximate fire frequency globally within mafic and ultramafic regions, we combined a generalized geologic map with bedrock domain classification[83] and GlobFire Database[84] within a geographic information system (QGIS 3.10.4). Total global population, including urban and rural areas, was visualized at 1-km resolution using a projection model for 2020 based on a "middle-of-the-road" socio-economic scenario (SSP2) (Supplementary Fig. 1b)[85,86].

Among other themes, the generalized geologic map designates a mafic-ultramafic igneous theme in which contains mafic and/or ultramafic igneous domains (or their metamorphic equivalents) that are classified as intrusive or extrusive and subdivided by era (Supplementary Fig. 1a). At the global scale, domains representing small outcrops of mafic and ultramafic rocks may be exaggerated in size while clusters of mafic-ultramafic domains appear as single polygon units. Geologic substrate can be further resolved by integrating regional and statewide geology maps (rather than a generalized global map) to better assess exposure risks of geogenic metals based on bedrock type.

The GlobFire Database (version 2, Jan. 2021) contains individual fire perimeters from 2001 to 2020 that are derived from the satellite MODIS burned area collection 6 product (MCD64A1). Fire events were overlaid on mafic-ultramafic geologic units and filtered by final area size. Large and small fires were respectively defined if the area equaled or exceeded and was less than approximately 2023 hectares, the size

distinction for Class G wildfires[87]. Importantly, devastating wildfires within WUIs range in fire sizes. Smaller fires in a densely populated WUI may pose a greater risk to PM exposure than expansive wildland fires in remote areas that burn at lower fire intensity; therefore, fire size does not simply correlate with exposure risks. Using a grid (5 × 5 km pixel resolution) within mafic-ultramafic regions, we calculated the frequency (count) of fire events within each grid pixel. Global maps are presented using the Robinson projection.

Fire regimes vary across ecosystem types, and the extent of exposure risks to extreme wildfires in certain regions cannot be completely captured within the available satellite-derived global fire database, particularly considering future shifts from climate change[7,88]. For example, many forest ecosystems (e.g., boreal, temperate) have fire frequencies on the decadal and century timescales. Savannah and grassland ecosystems burn at a higher frequency (1-5 years) but at a lower fire intensity and thus severity because of smaller fuel loads. Regional trend assessments of fire activity, where historical fire records extend earlier than 2001, are critical for understanding and predicting fire frequency and severity[7].

## Reporting summary

Further information on research design is available in the Nature Portfolio Reporting Summary linked to this article.

## Data availability

Data supporting the findings of this study are available within the article, its Supplementary Information file, and source data have been deposited in the Stanford Digital Repository at https://purl.stanford.edu/gj966nd5234. Geospatial data are publicly available online. California regional geologic maps are available at the United States Geological Survey (USGS) (https://pubs.er.usgs.gov). Vegetation data for Sonoma County are available at https://sonomavegmap.org. California soil data are available at https://websoilsurvey.nrcs.usda.gov. Global fire perimeters (2001-2020) are available at the Global Wildfire Information System (GWIS) (https://gwis.jrc.ec.europa.eu). A generalized global geologic map is available at https://geoscan.nrcan.gc.ca. Global population projection grids are available at NASA Socioeconomic Data and Applications Center (SEDAC) (https://beta.sedac.ciesin.columbia.edu).

## Code availability

The R code used for statistical analyses has been deposited in the Stanford Digital Repository at https://purl.stanford.edu/gj966nd5234.

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

## Acknowledgements

This project was supported by Stanford University's Vice Provost for Graduate Education (EDGE Fellowship) (A.M.L.), Doerr School of Sustainability (McGee Levorsen Grant) (A.M.L.), and UPS Foundation Endowment Funds (A.M.L., S.F.). We thank the staff at Pepperwood

Foundation (PP), Audubon Canyon Ranch (ACR), and the Sonoma Land Trust (SLT) for collaboration and logistical support in performing fieldwork at Pepperwood, Modini, and White Rock Preserves, respectively. This work was performed in part at the University of California Natural Reserve System, McLaughlin Natural Reserve (MLNR) DOI: 10.21973/N3W08D. We thank our colleagues W. Flower, A. Duncan, E. Paulus (Stanford), J. Bradbury (ACR), R. Ferrell (PP), and C. Koehler (MLNR) for their fieldwork support, B. Melosh (USGS) for geologic expertise at MLNR, as well as A. Gomes, M. Capetz, D. Burns, D. Turner, and G. Li (Stanford), S. Bone (SSRL), and A. Foster and K. Perkins (USGS) for their laboratory and analytical support. We are grateful for feedback from R. Ferrell, M. Halbur, T. Commendant (PP), M. Cooper (ACR), C. Koehler (MLNR), and M. Hammar (SLT) on our manuscript pre-submission. We also appreciate discussions with Prof. Ed Burton on the generation and threat of metals within fire impacted soils. Use of the Stanford Synchrotron Radiation Lightsource, SLAC National Accelerator Laboratory, is supported by the U.S. Department of Energy, Office of Science, Office of Basic Energy Sciences under Contract No. DE-AC02-76SF00515. The SSRL Structural Molecular Biology Program is supported by the DOE Office of Biological and Environmental Research, and by the National Institutes of Health, National Institute of General Medical Sciences (P30GM133894). The contents of this publication are solely the responsibility of the authors and do not necessarily represent the official views of NIGMS or NIH. Publicly available data associated with the Sonoma Vegetation and Habitat Map was supported by NASA Grant NNX13AP69G, the University of Maryland, and the Sonoma Vegetation Mapping and LiDAR Program.

## Author contributions

A.M.L. and S.F. designed the study. A.M.L. organized and conducted fieldwork, collected data, and performed analyses. J.L.P. helped to generate and analyze X-ray data. A.M.L. led the writing of the manuscript with regular input from S.F. All authors commented on the manuscript and gave approval for publication.

## Competing interests

The authors declare no competing interests.
