## [Peer Review File · Nature Communications]

Metal Toxin Threat in Wildland Fires Determined by Geology and Fire SeverityREVIEWER COMMENTS

Reviewer #1 (Remarks to the Author):

Review of manuscript 395109 submitted to Nature Communication

Metal toxin threat in wildland fires determined by geology and fire severity

Alandra Marie Lopez, Juan Lezama Pacheco and Scott Fendorf

This manuscript presents the results of a study aimed at emphasizing the health threat potentially arising from chromium transformation to its most harmful Cr(VI) form in soils and ashes upon wildfires. To reach their objective, the authors analyzed 38 cores drilled in more or less severely burned soils at four Preserves that experienced large wildfires in the North Coast Range of California (USA). The results obtained confirm the catalytic effect of high temperatures generated by wildfires on chromium oxidation in soil, and they point to control of soil geology and fire severity on this effect. They also interestingly indicate that reactive Cr(VI) can reach dangerous levels in wind-dispersible particulates found in the surficial layers of the soil at ultramafic settings, and that this harmful form of chromium can persist in the soil/ash system up to one year after the wildfire if rainfalls are not significant. These results yield to the conclusion that more work is required to further evaluate this potential risk around the world.

I found this paper very pleasant to read. The subject is well introduced, the sites and samples are well described and fit with the objective of the study, the results are well presented and they support the discussion that gives proper attention to the existing literature on the topic and yield to the novel conclusions that geology and fire intensity are important drivers of harmful hexavalent chromium in soils and wind-dispersible soil/ash surface particles, and that the associated health risk can persist up to one year after fire.

Although the scale of the study (North Coast Range of California, USA) could at first be considered too short to extend its findings at the global scale, I agree with the authors that the number and diversity of sites studied can be considered enough to address this issue. I also agree with the concluding remark on the importance of further evaluating the potential risk of wildfire-induced harmful hexavalent chromium in wind-dispersible soil/ash particles at the global scale. Finally, I agree with the authors that the results provided in this study deserve to be shared with a large audience, ranging from scientists working on the topic to policy makers, public administrators and nature managers.

For all these reasons, I consider that this manuscript deserves to be published in *Nature Communications*. I have listed below few issues that I would be interested to see addressed, although only few of them are mandatory for publication.

Figure 1

I agree that a large fraction of ultramafic/mafic areas are concerned by wildfires at the global scale. Figure 1 shows that these areas are mainly located in the tropical region where most soils are deeply weathered (Ultisols and Oxisols, according to the USDA classification). However, the Fe contents reported in Table S6 suggest that the soils studied in this work do not correspond to these soil types. This raises the question of the actual representativeness of the results regarding tropical areas. I would

recommend that the authors comment on that point, and maybe further consider it in their concluding remarks.

Figure 3

Maybe, the authors could add the results of comparative statistics tests on barplots and boxplots (either directly report the p-values or show corresponding labels as *** or **) ?

Do they have any hypothesis to explain the much higher concentration in reactive Cr(VI) measured in the A7 surface soil-ash sample from the serpentine chaparral landscape ? The data provided in Table S6 indicate about twice more total Cr in this sample compared to sample A6 (for instance), but the concentration in reactive Cr(VI) is more than 3 times higher in the bulk fraction and more than 25 three times higher in the <53 μ m fraction. Are there any mineralogical differences with the other moderately-highly burned serpentine soil-ash samples (A3-A6) that could help to explain that ?

Figure 4

I wonder if this figure should be maintained in the main text. First, I am not convince that it really supports the assumption that “*total Cr(VI) was most abundant in wind-dispersible soils and ash particulates after high fire severity conditions compared to a low severity sample*”. Second, I do not understand why the two figures are plotted on different sizes (A is larger than B).

It could maybe be replaced by a figure similar to Figure S7 (maybe use only panels A and B or panels C and D from Figure S7 and add two similar panels from a low fire severity burn sample from a serpentine chaparral soil) ?

Whatever, if Figure 4 is maintained in the main text, I would recommend that the authors add some histograms with the estimated number of Cr(VI) and total Cr particles on the two figures, in order to help the reader to better assess the relative proportion of both types of particles. At least, I would recommend that they change for more contrasted colors, in order to help the reader to better visually decipher between Cr(VI) and total Cr particles.

Figure S6

I would recommend to change the letters for A, B and C in the legend to fit with the letters reported on the figure.

Table S6

Total concentrations in bulk surface soil-ash samples (and some <53 μ m fractions) are displayed, but total concentration in bulk soil samples across the whole soil cores are not provided. I would recommend that the authors provide these data (at least those for Cr) as mean total concentration for rhyolite, mélange and serpentine soils in SI, either in the form of a Table or as a figure similar to Figure 2. Such a figure would better show that the fractions of reactive Cr(VI) is very low compared to total Cr concentration. In the same way, I would have been interested to see a figure similar to Figure 2 that would have depicted the fraction of reactive Cr(VI) as a function of the total Cr concentration. Indeed, such a figure would have helped to check if the fraction of reactive Cr(VI) is really higher in serpentine soils. Even if I agree with the authors that the concentration of reactive Cr(VI) is the most relevant parameter to assess a potential environmental and/or health risk, the fraction of reactive Cr(VI) could further inform on the actual mechanism(s) and/or soil characteristic(s) that favor Cr(III) to

Cr(VI) oxidation in burned soils. But, maybe this question is beyond the scope of the paper...

Figure S7

Legend : ... from a serpentine chaparral soil...

Why did the authors not tried to analyze more some Cr(VI) areas on panel A ?

The XANES spectrum at point 1 on panel C shows a well-marked Cr(VI) pre-edge peak but the color code indicate rather low amounts of Cr(VI) at this point. Could the authors explain that ?

Suggested additional references

Rascio et al., 2022. Evidence of hexavalent chromium formation and changes of Cr speciation after laboratory-simulated fires of composted tannery sludges long-term amended agricultural soils. *Journal of Hazardous Materials*, 436, 129117. <https://doi.org/10.1016/j.jhazmat.2022.129117>

Terzano et al., 2021. Fire effects on the distribution and bioavailability of potentially toxic elements (PTEs) in agricultural soils. *Chemosphere*, 130752. <https://doi.org/10.1016/j.chemosphere.2021.130752>

Ré et al., 2021. Cytotoxic effects of wildfires ashes : *In-vitro* responses of skin cells. *Environmental Pollution*, 285, 117279. <https://doi.org/10.1016/j.envpol.2021.117279>

Jahn et al., 2021. Metallic and crustal elements in biomass-burning aerosols and ash: Prevalence, significance, and similarities to soil particles. *ACS Earth and Space Chemistry*, 5, 136-148. <https://dx.doi.org/10.1021/acsearthspacechem.0c00191>

Xu et al., 2020. Wildfires, global climate change, and human health. *The New England Journal of Medicine*, 383, 2173-2181. <https://doi.org/10.1056/NEJMSr2028985>

Reviewer #2 (Remarks to the Author):

The authors have to highlight the novelty of their manuscript. The abstract should be revised to attract the reader's attention. The Introduction section should be improved by adding references dealing soil contamination issues. However, the problem is that the English and the whole organisation of the present version are definitely below an acceptable standard for an international scientific journal. Analytical quality control is missing. Detection limits of the applied methods should be reported. The main problem for this manuscript is its structure. Major parts are missing from the manuscript. My suggestion is to reject this manuscript and encourage the authors to submit a more mature manuscript.

Response to Review Comments

Reviewer #1

This manuscript presents the results of a study aimed at emphasizing the health threat potentially arising from chromium transformation to its most harmful Cr(VI) form in soils and ashes upon wildfires. To reach their objective, the authors analyzed 38 cores drilled in more or less severely burned soils at four Preserves that experienced large wildfires in the North Coast Range of California (USA). The results obtained confirm the catalytic effect of high temperatures generated by wildfires on chromium oxidation in soil, and they point to control of soil geology and fire severity on this effect. They also interestingly indicate that reactive Cr(VI) can reach dangerous levels in wind- dispersible particulates found in the surficial layers of the soil at ultramafic settings, and that this harmful form of chromium can persist in the soil/ash system up to one year after the wildfire if rainfalls are not significant. These results yield to the conclusion that more work is required to further evaluate this potential risk around the world.

I found this paper very pleasant to read. The subject is well introduced, the sites and samples are well described and fit with the objective of the study, the results are well presented and they support the discussion that gives proper attention to the existing literature on the topic and yield to the novel conclusions that geology and fire intensity are important drivers of harmful hexavalent chromium in soils and wind-dispersible soil/ash surface particles, and that the associated health risk can persist up to one year after fire.

Although the scale of the study (North Coast Range of California, USA) could at first be considered too short to extend its findings at the global scale, I agree with the authors that the number and diversity of sites studied can be considered enough to address this issue. I also agree with the concluding remark on the importance of further evaluating the potential risk of wildfire-induced harmful hexavalent chromium in wind- dispersible soil/ash particles at the global scale. Finally, I agree with the authors that the results provided in this study deserve to be shared with a large audience, ranging from scientists working on the topic to policy makers, public administrators and nature managers.

For all these reasons, I consider that this manuscript deserves to be published in Nature Communications. I have listed below few issues that I would be interested to see addressed, although only few of them are mandatory for publication.

Response: We thank the reviewer for their constructive feedback and review of the manuscript's findings and implications. We have reviewed and addressed in detail below the reviewer's suggestions.

Changes: Please see our specific changes to the reviewer's suggestions below.

Figure 1: *I agree that a large fraction of ultramafic/mafic areas are concerned by wildfires at the global scale. Figure 1 shows that these areas are mainly located in the tropical region where most soils are deeply weathered (Ultisols and Oxisols, according to the USDA classification). However, the Fe contents reported in Table S6 suggest that the soils studied in this work do not*

correspond to these soil types. This raises the question of the actual representativeness of the results regarding tropical areas. I would recommend that the authors comment on that point, and maybe further consider it in their concluding remarks.

Response: We agree with the reviewer that the soils in this study are not as deeply weathered as Oxisols and Ultisols, like lateritic soils, common across tropical climate regions (e.g., New Caledonia, Cuba, Brazil, Malaysia, Indonesia, Madagascar, northern Australia). In our study, the average Fe concentration across burned and unburned bulk ultramafic (serpentine) soil was 8.71 wt. %, while lateritic soils can contain more than 5x more Fe, typically in the form of crystalline Fe oxides (e.g., hematite, goethite). Chromium is predominantly found within these crystalline Fe oxides. Additionally, lateritic soils are often depleted in Ca, Mg, and Si. The Fe concentrations of our study soils are representative of serpentine soils in Mediterranean or temperate climates (e.g., California, Oregon, Washington, Turkey, Balkans) with moderate Fe oxide content and neutral to alkaline pH. We have added a paragraph within the *Discussion* section that addresses the characterization of ultramafic soils globally and the limitations of soil types within our study site when considering mechanisms in lateritic and highly weathered metal-rich soils in the tropical regions. Furthermore, we place our findings from field analysis in the context of laboratory studies that examine Cr(VI) generation from Cr(III) solids common to lateritic soils.

Changes: Within the *Discussion* we now state:

“While recognized for urban fires, threats from metal exposure in smoke and dust need to be recognized within wildland fires arising on metal-rich geologies. Across tropical climate regions, deeply weathered lateritic soils are common, in which Cr is predominantly found within crystalline Fe oxides (e.g., hematite, goethite), and soil Fe content may exceed 50 wt. %⁶². Past work has quantified the Cr oxidation capacity of Cr-bearing Fe oxides and lateritic soils during heating simulations^{35,62}. For example, the greatest Cr(VI) formation (more than 40% of total Cr) upon heating hematite occurred at temperatures less than 400°C, while up to 100% of total Cr in goethite transformed to Cr(VI) at 800°C. In Mediterranean or temperate climates (similar to this study’s region), ultramafic soils are relatively more enriched in Cr-bearing phyllosilicate minerals (e.g., serpentine), Fe oxide content is moderate (typically less than 10 wt. %) with more amorphous secondary phases, and soil pH is neutral to alkaline pH. In our study, and under natural wildfire conditions, we observed that up to about 0.015% of total Cr was reactive Cr(VI) in burned serpentine soils. Chromium(VI) generation during wildfires depend on fire conditions and host mineralogy; thus, the extent of Cr(VI) formation in lateritic soils may differ from temperate serpentine soils, but field observations of Cr(VI) in burned lateritic soils following wildfires are currently lacking.”

Figure 3: Maybe, the authors could add the results of comparative statistics tests on barplots and boxplots (either directly report the p-values or show corresponding labels as *** or **)?

Response: We performed a one-way ANOVA for Figure 3c and found no statistically significant difference between the three groups (p-value = 0.164). In Figure 3b, we are illustrate reactive concentrations within surface soil-ash based on fire severity (low

versus moderate-high) and particle size (bulk soil less than 2 mm versus the silt and clay-sized fraction less than 53 μm). We are unable to run a two-way ANOVA because burn severity sample sizes are not equal.

Changes: For Figure 3, we now state "Reactive Cr(VI) concentrations ranged from 64 to 1,060 $\mu\text{g}/\text{kg}$, with a median concentration of 257 $\mu\text{g}/\text{kg}$, and remained elevated compared to concentrations (5-64 $\mu\text{g}/\text{kg}$) within the near surface depths (0-2 cm) of unburned serpentine soil (Figure 3c); however, these differences were not statistically significant ($p\text{-value} = 0.164$). "

Do they have any hypothesis to explain the much higher concentration in reactive C(VI) measured in the A7 surface soil-ash sample from the serpentine chaparral landscape? The data provided in Table S6 indicate about twice more total Cr in this sample compared to sample A6 (for instance), but the concentration in reactive Cr(VI) is more than 3 times higher in the bulk fraction and more than 25 three times higher in the $<53\mu\text{m}$ fraction. Are there any mineralogical differences with the other moderately- highly burned serpentine soil-ash samples (A3-A6) that could help to explain that?

Response: At site A7, the soil experienced longer burning duration and fire intensities with greater biomass combustion that may further contribute to the high-levels of Cr(VI). Unlike A7 (original Figure S9; revised Figure S11), we did not observe mineralogical changes in bulk composition for samples A3-A6 in the surface soil and ash compared to underlying burned soil. We suspect that high temperatures did not persist for sufficient time to alter bulk mineralogy in the latter samples.

Changes: Within the *Results* we state:

"At site A7, the average reactive Cr(VI) concentration was more than three times greater than other moderate-high fire severity sites (Figure 3b). We suspect that longer burning duration and fire intensities with greater biomass combustion contributed to the relatively high-levels of Cr(VI), as this was a severely burned forested area. Importantly, ash from severely burned areas concentrate alkali (Na, K) and alkaline earth (Ca, Mg) metals (often from biomass combustion) that are key for the thermal oxidation of Cr(III)^{13,43}. For example, CaCrO_4 was noted after agricultural soil amended with composted Cr(III)-rich tannery sludge was heated at 500°C⁵⁵."

Figure 4: *I wonder if this figure should be maintained in the main text. First, I am not convince that it really supports the assumption that 'total Cr(VI) was most abundant in wind- dispersible soils and ash particulates after high fire severity conditions compared to a low severity sample'. Second, I do not understand why the two figures are plotted on different sizes (A is larger than B).*

It could maybe be replaced by a figure similar to Figure S7 (maybe use only panels A and B or panels C and D from Figure S7 and add two similar panels from a low fire severity burn sample from a serpentine chaparral soil)?

Whatever, if Figure 4 is maintained in the main text, I would recommend that the authors add some histograms with the estimated number of Cr(VI) and total Cr particles on the two figures, in order to help the reader to better assess the relative proportion of both types of particles. At least, I would recommend that they change for more contrasted colors, in order to help the reader to better visually decipher between Cr(VI) and total Cr particles.

Response: We appreciate the reviewer's feedback regarding Figure 4. We agree that our results of greater Cr(VI)-bearing particles in high severity soil-ash versus low-severity conditions can be clearer, especially by including a histogram of particles containing Cr(VI). Regarding the reviewer's concern about the different size plots, the sample area for large-scale micro-XRF maps of each thin section were not held constant during data collection, resulting in different mapped areas. Due to limitations of the analysis software, we are unable to change for more contrasted colors, but we can update the min/max values for color brightness.

We have revised Figure 4 to focus on a 1- μ m resolution XRF image containing particles from a high fire severity sample with and without measurable Cr(VI) by XANES analysis. We then moved the original Figure 4 to the Supplementary Information as Figure S8, and added another example of Cr(VI)-containing particles in a 1- μ m resolution XRF image as Figure S9. We also revised the main text related to the figures to highlight the presence of Cr(VI)-containing particles within the high fire severity samples, which was not apparent in low fire severity samples.

Changes: See revised Figure 4, Figure S8, and Figure S9 below.

“Using micro-scale X-ray techniques, Cr(VI)-containing soil and ash particulates were identified in a high fire severity sample as opposed to a low fire severity sample (**Figure 4; Figure S8**). Here, Cr(VI) was associated with mineral surfaces (e.g., adsorbed) or enriched in relatively low-Cr particles with Ca and K (**Figure 4, Figure S9**). Consistent with particle analysis, reactive Cr(VI) concentrations spanned from 326 to 13,000 μ g/kg (**Figure 3b**).”

Figure 4 | Total Cr(VI) in wind-dispersible soil and ash particles. a. μ -XRF image (pixel resolution: 1 μ m) showing the relative intensity of Cr(VI) (green; estimated as the intensity ratio at 5993 and 6010 eV) and total Cr (blue; measured at 6010 eV) within the < 53- μ m size fraction of Cr-bearing soil-ash particulates from a serpentine chaparral that experienced high fire severity (A7). **b.** Normalized μ -XANES spectra (Cr K-edge) from numbered locations on Cr-bearing particles in **a**. Dashed lines indicate energies characteristic of Cr(VI) (5993 eV), Cr(III) (6003 eV), and total Cr (6010 eV), at which μ -XRF images were also collected.

Figure S8 | Total Cr(VI) in wind-dispersible soil and ash particles. μ -XRF image showing particle distribution of total Cr(VI) (green; estimated as the intensity ratio at 5993 and 6010 eV) and total Cr (blue; measured at 6010 eV) within the < 53- μ m size fraction of Cr-bearing soil-ash particulates from **a.** high fire severity site (A7) and **b.** low fire severity site (A1) in a serpentine chaparral.

Figure S9 | Total Cr(VI) in wind-dispersible soil and ash particles. **a.** μ -XRF image (pixel resolution: 1 μ m) showing the relative intensity of Cr(VI) (green; estimated as the intensity ratio at 5993 and 6010 eV) and total Cr (blue; measured at 6010 eV) within the < 53- μ m size fraction of Cr-bearing soil-ash particulates from a serpentine chaparral that experienced high fire severity (A7). **b.** Normalized μ -XANES spectra (Cr K-edge) from numbered locations on Cr-bearing particles in **a.** Dashed lines indicate energies characteristic of Cr(VI) (5993 eV), Cr(III) (6003 eV), and total Cr (6010 eV).

Figure S6: *I would recommend to change the letters for A, B and C in the legend to fit with the letters reported on the figure.*

Response: Based on the reviewer's recommendation, we have revised the figure, accordingly, by changing uppercase letters to lowercase, in addition to the other figures with sub-panels (similarly identified by Reviewer #2).

Changes: We changed the letters to lowercase, as requested.

Table S6: *Total concentrations in bulk surface soil-ash samples (and some <53 μm fractions) are displayed, but total concentration in bulk soil samples across the whole soil cores are not provided. I would recommend that the authors provide these data (at least those for Cr) as mean total concentration for rhyolite, m \acute{e} lange and serpentine soils in SI, either in the form of a Table or as a figure similar to Figure 2. Such a figure would better show that the fractions of reactive Cr(VI) is very low compared to total Cr concentration.*

Response: We thank the reviewer for their recommendation, which we have addressed in the revision. Please refer to our response and changes to the reviewer's next comment related to including a figure showing the fraction of reactive Cr(VI) to total Cr concentrations.

Changes: Using total Cr concentrations reported for soil cores in Table S1, we have revised **Table S6** to include the mean total element concentrations for Cr, Fe, Mn, Ni, Ca, Mg, Na, and K in addition to the surface soil-ash sample data so that the reader can compare elemental concentrations in surface soil and ash to bulk soil from different geologies.

Table S6 | Physicochemical characteristics of bulk soil and ash (up to 2 mm), and selected fine size fractions less than 53 μm , collected from surface layers of the burned serpentine chaparral, and mean elemental concentrations from bulk underlying soil based on geology type (rhyolitic, mélange, and serpentine).

ID	Fire Severity ^a	% Sand ^b (2-0.05 mm)	% Silt ^c (53-2 μm)	% Clay ^c (< 2 μm)	Cr mg/kg	Fe mg/g	Mn mg/kg	Ni mg/kg	Ca mg/g	Mg mg/g	Na mg/g	K mg/g
Surface Soil-Ash												
A1	L				1147	64.7	1351	1528	15.5	77.3	4.63	3.76
A2	L				1532	69.1	1203	2530	10.2	158	< 0.1	1.65
A3	M/H				1606	84.2	1480	3117	9.3	148	< 0.1	1.54
A4	M/H	87.5	11.7	0.8	2256	78.9	1438	3380	8.7	159	< 0.1	0.90
A5	M/H				999	57.7	1022	1849	10.2	125	2.10	2.48
A6	M/H	78.2	20.2	1.6	1970	79.2	1555	2726	29.9	174	< 0.1	2.86
A7	M/H	85.7	12.9	1.4	4829	102	1543	2643	17.9	211	< 0.1	2.29
Less than 53 μm size fraction												
A4	M/H				1133	94.7	1975	3042	41.5	124	< 0.1	3.38
A6	M/H				946	82.7	1910	2294	44.2	133	< 0.1	4.41
A7	M/H				1643	105	2547	3489	34.0	189	2.02	3.91
Bulk Soil^d												
	Rhyolitic (n = 7)				162	34.5	796	88	10.7	7.9	13.6	9.70
	Mélange (n = 16)				314	50.0	855	259	10.7	37.3	8.17	13.2
	Serpentine (n = 10)				2373	87.1	1511	2929	4.4	150	3.61	1.69

^a L = Low severity, M = Moderate severity, H = High severity

^b Determined by sieve analysis

^c Determined by laser diffraction particle size counter

^d Bulk soil concentrations are mean values using all soil cores (fire-affected and unburned) for each geology type: rhyolitic (n = 7), mélange (n = 16), and serpentine (n = 10).

In the same way, I would have been interested to see a figure similar to Figure 2 that would have depicted the fraction of reactive Cr(VI) as a function of the total Cr concentration. Indeed, such a figure would have helped to check if the fraction of reactive Cr(VI) is really higher in serpentine soils. Even if I agree with the authors that the concentration of reactive Cr(VI) is the most relevant parameter to assess a potential environmental and/or health risk, the fraction of reactive Cr(VI) could further inform on the actual mechanism(s) and/or soil characteristic(s) that favor Cr(III) to Cr(VI) oxidation in burned soils. But, maybe this question is beyond the scope of the paper...

Response: We thank the reviewer for their recommendation. We agree that the reactive Cr(VI) fraction of total Cr in soil and soil-ash is low; however, the fraction is relatively higher in near surface soil depths compared to past studies quantifying natural Cr(III) oxidation in unburned soils, including a 2017 study at McLaughlin Natural Reserve. Moreover, as the reviewer notes, the hazard imposed by the particulates is related to the reactive Cr(VI). The percentage of total Cr that was reactive Cr(VI) in unburned serpentine soil was consistent with previous measurements (McClain et al., 2017). Interestingly, the reactive Cr(VI) fraction differs based on geology (rhyolite, mélange, and serpentinite). The reactive Cr(VI) fraction in rhyolitic and mélange soils composed more of the total Cr content than the relative fraction within serpentine soils. In order to highlight these variations across soil depth in fire-affected and unburned sites, we added a figure to the supplementary information.

Changes: **Figure S6** (below) was added to the Supplementary Information. Succeeding figure numbers were updated based on this addition. Within the *Results* and *Discussion* sections, we now state:

“Average Cr(VI) concentrations generated in soils derived from mélange (**Figure 2b**) were more than double the respective levels in rhyolitic soil (**Figure 2a**) regardless of overlapping ranges in total Cr content, 152-954 and 102-338 mg/kg, respectively, reflecting the potential contribution of differing mineralogy to Cr(VI) generation (**Figure S6**)³⁵.”

“In our study, and under natural wildfire conditions, we observed that up to about 0.015% of total Cr was reactive Cr(VI) in burned serpentine soils (**Figure S6**).”

Figure S6 | Fraction of total Cr that is reactive Cr(VI) in burned and unburned soils. The ratio of reactive Cr(VI) to total Cr concentrations (as a percentage) within **a.** rhyolite-, **b.** mélange-, **c.** serpentinite-derived soil profiles (0-16 cm) that were not burned (gray; rhyolite, $n = 3$; mélange, $n = 7$; serpentinite, $n = 3$) or were fire-affected (colored; rhyolite, $n = 4$; mélange, $n = 9$; serpentinite, $n = 7$). Percentages were also plotted for serpentinite-derived soil (0-20 cm) from McClain *et al.* (2017) in **c.** for comparison. Each point represents the average percentage for a soil core based on triplicate measurements.

Figure S7: Legend : ... from a serpentine chaparral soil...

Response: We agree with the reviewer’s recommendation and have made the appropriate changes. Based on the reviewer’s previous recommendations, this figure has revised as Figure 4 and Figure S9.

Changes: (Figure 4 and Figure S9 caption) "... particles (< 53 μm) from a serpentine chaparral that experienced high fire severity ..."

Why did the authors not tried to analyze more some Cr(VI) areas on panel A ?

The XANES spectrum at point 1 on panel C shows a well-marked Cr(VI) pre-edge peak but the color code indicate rather low amounts of Cr(VI) at this point. Could the authors explain that ?

Response: Our XANES analysis was used to corroborate our bulk measurements of reactive Cr(VI). Particles within the XRF map were used to denote the presence and abundance of Cr(VI), corroborating (and visualizing) the reactive fraction measurements.

Changes: “To corroborate bulk measurements of reactive Cr(VI), we combined multi-energy mapping with μ -XANES of select spots on particles to confirm the presence of Cr(VI).”

Suggested additional references

Rascio *et al.*, 2022. Evidence of hexavalent chromium formation and changes of Cr speciation after laboratory-simulated fires of composted tannery sludges long-term amended agricultural

soils. *Journal of Hazardous Materials*, 436, 129117.

<https://doi.org/10.1016/j.jhazmat.2022.129117>

Terzano et al., 2021. Fire effects on the distribution and bioavailability of potentially toxic elements (PTEs) in agricultural soils. *Chemosphere*, 130752.

<https://doi.org/10.1016/j.chemosphere.2021.130752>

Ré et al., 2021. Cytotoxic effects of wildfires ashes : In-vitro responses of skin cells.

Environmental Pollution, 285, 117279. <https://doi.org/10.1016/j.envpol.2021.117279>

Jahn et al., 2021. Metallic and crustal elements in biomass-burning aerosols and ash:

Prevalence, significance, and similarities to soil particles. *ACS Earth and Space Chemistry*, 5, 136-148. <https://dx.doi.org/10.1021/acsearthspacechem.0c00191>

Xu et al., 2020. Wildfires, global climate change, and human health. *The New England Journal of Medicine*, 383, 2173-2181. <https://doi.org/10.1056/NEJMs2028985>

Response: We thank the reviewer for sharing additional references. We agree that these references are relevant to the study.

Changes: We have added these references suggested above, in addition to a few other recently published and relevant studies, to discussions within the main text.

“Global wildfire activity represents a rising distributed health risk from smoke and dust inhalation⁵⁻¹⁰.”

10. Xu, R. et al. Wildfires, Global Climate Change, and Human Health. *N. Engl. J. Med.* **383**, 2173–2181 (2020).

“Increased heavy metals in PM have been documented during wildfire episodes and may induce cytotoxicity, increase lung cancer risks, and greatly contribute to oxidative stress¹⁹⁻³⁰.”

“Suburban fires illustrates the impacts of inhaling Cr(VI)-containing ash within the respiratory tract by measuring Cr(VI) leached with a simulated lung fluid^{36,37} and discerning Cr mineralogy within nano-sized particulates (< 100 nm)^{22,30}.”

21. Boaggio, K. et al. Beyond Particulate Matter Mass: Heightened Levels of Lead and Other Pollutants Associated with Destructive Fire Events in California. *Environ. Sci. Technol.* **56**, 14272–14283 (2022).

22. Alshehri, T. et al. Wildland-urban interface fire ashes as a major source of incidental nanomaterials. *J. Hazard. Mater.* **443**, 130311 (2023).

28. Jahn, L. G. et al. Metallic and crustal elements in biomass-burning aerosol and ash: Prevalence, significance, and similarity to soil particles. *ACS Earth Sp. Chem.* **5**, 136–148 (2021).

29. Ré, A. et al. Cytotoxic effects of wildfire ashes: In-vitro responses of skin cells. *Environ. Pollut.* **285**, 117279 (2021).

“Metals in soils and ash are commonly linked to structural burning within wildland-urban interfaces (WUI) ^{1,31-33}, with negligible awareness of wildland landscapes (soils and ash) as an alternative and highly distributed source ¹.”

32. Alexakis, D. E. Suburban areas in flames: Dispersion of potentially toxic elements from burned vegetation and buildings. Estimation of the associated ecological and human health risk. *Environ. Res.* **183**, 109153 (2020).

33. Alam, M. et al. Identification and quantification of Cr, Cu, and As incidental nanomaterials derived from CCA-treated wood in wildland-urban interface fire ashes. *J. Hazard. Mater.* **445**, 130608 (2023).

“Following wildfires, severely burned areas are often barren and blanketed with ash and loose, rough topsoil leading to enhanced post-fire wind and water erosion ^{8,39-42}.”

41. Yu, Y. & Ginoux, P. Enhanced dust emission following large wildfires due to vegetation disturbance. *Nat. Geosci.* 2022 *1511* **15**, 878–884 (2022).

42. Shakesby, R. A. & Doerr, S. H. Wildfire as a hydrological and geomorphological agent. *Earth-Science Rev.* **74**, 269–307 (2006).

“For example, CaCrO_4 was noted after agricultural soil amended with composted Cr(III)-rich tannery sludge was heated at 500°C ⁵⁵.”

55. Rascio, I. et al. Evidence of hexavalent chromium formation and changes of Cr speciation after laboratory-simulated fires of composted tannery sludges long-term amended agricultural soils. *J. Hazard. Mater.* **436**, 129117 (2022).

Reviewer 2

The authors have to highlight the novelty of their manuscript. The abstract should be revised to attract the reader's attention. The Introduction section should be improved by adding references dealing soil contamination issues. However, the problem is that the English and the whole organisation of the present version are definitely below an acceptable standard for an international scientific journal. Analytical quality control is missing. Detection limits of the applied methods should be reported. The main problem for this manuscript is its structure. Major parts are missing from the manuscript. My suggestion is to reject this manuscript and encourage the authors to submit a more mature manuscript.

Response: We appreciate the reviewer's feedback and have sought to make the abstract, and the manuscript, have more pizzazz. With that said, the manuscript was formatted specifically to the Nature guidelines. Further, the authors are all native speakers, and the senior author has published several hundred articles, including ones in *Nature* and *Science*. The present manuscript holds to those same standards. It is also worth noting that counter to Reviewer 2, Reviewer 1 stated "I found this paper very pleasant to read. The subject is well introduced, the sites and samples are well described and fit with the objective of the study, the results are well presented and they support the discussion...". Further, several established authors at Stanford have read the manuscript and all support the writing and presentation. Thus, while we don't want to dismiss the comments of the Reviewer, we do see them as anonymous in comparison to others.

Based on the reviewer's feedback, we have revised our *Methods* section to include more information regarding our analyses. We have added detection limits for Cr(VI) and NH₄ measurements, and have revised related Figures and Tables to reflect these changes. With the exception of Na (detection limit of 0.1 mg/g), total concentrations for all elements reported using XRF were significantly greater than respective detection limits. We periodically analyzed certified reference material, NIST 2711a, with bulk soil samples to confirm accuracy of the XRF instrument. For aqueous extractions and associated instrument analyses, we tracked quality assurance in multiple ways. For each round of aqueous extractions using 10 mM K₂HPO₄/KH₂PO₄ solution, we included at least two centrifuge tubes containing the phosphate buffer solution and no soil/ash that were analyzed similar to samples for Cr(VI) and NH₄. Unless soil mass was limited, extractions were conducted in triplicate to assess sample heterogeneity. On the UV-Vis and ICP-MS, we analyzed instrument blanks every 15-20 samples and multiple quality control standard solutions prepared with certified Cr reference solutions throughout each analysis.

Changes: We have modified the *Abstract*, added references to the *Introduction* section, and have sought to ensure the format and writing are consistent with the expected quality of the *Nature* journals. We have also revised the *Methods* to include detection limits, where relevant, and have similarly revised Figures and Tables to reflect non-detectable sample concentrations throughout the manuscript.

Within the *Methods* section, we now state:

"Aqueous Extractions and Chemical Analysis

Reactive Cr(VI) concentrations (most available fraction, including dissolved and adsorbed Cr(VI)) in ~~bulk~~ soil ash samples (bulk and particle size fraction < 53 μm) and within soil cores were extracted with 10 mM $\text{K}_2\text{HPO}_4/\text{KH}_2\text{PO}_4$ (buffered at pH 7.2) ^{79,80}. Phosphate effectively competes with Cr(VI) ions for surface adsorption sites. At circumneutral to alkaline pH ranges in natural soils, it's expected that nearly all aqueous Cr is present in the hexavalent form, and that Cr(VI) concentrations will be primarily limited by adsorption ⁸¹. The clay size fraction (less than 2- μm diameter) typically has a dominant influence on species retention given their high surface areas and greater number of adsorption sites; therefore, it is likely that the reactive Cr(VI) concentrations measured here largely represent the fraction of Cr(VI) associated with clay particles. Triplicate samples were agitated in a 1:4 soil/solution ratio for 24 h, centrifuged (30 min, 4000 rpm, 4°C), and filtered through 0.22- μm filters. A subsample of unacidified filtrate was used to quantify aqueous Cr(VI) concentrations using the diphenylcarbazide (DPC) method on a UV-Vis spectrophotometer (Shimadzu UV-1601) ^{79,82}. The detection limit was 3 $\mu\text{g/L}$ (approximately 12 $\mu\text{g/kg}$) ⁸². Total Cr concentrations were determined with inductively coupled plasma mass spectrometry (ICP-MS, Thermo Scientific XSERIES 2), and confirmed that approximately all aqueous Cr was in the form of Cr(VI) in unburned soil and burned soil and ash, similarly observed in previous studies ^{36,37}. An aliquot of each soil extract was immediately acidified post-filtration and stored in 2% nitric acid at 4°C until ICP-MS analysis.

To determine relative differences in K^+ -extractable NH_4^+ concentrations (mg $\text{NH}_4^+\text{-N/kg}$) within burned and unburned soils (**Figure S10**), additional unacidified samples (after $\text{K}_2\text{HPO}_4/\text{KH}_2\text{PO}_4$ extraction) from 30 of the 38 total soil cores (21 fire-affected and 9 unburned soil cores) were frozen at -20°C until chemical analysis. Ammonium is a direct combustion product and will be elevated in the near surface soil after wildfires depending on burn severity ⁴⁰. Ammonium concentrations in the top 6-cm were measured in triplicate (when sample volume allowed) using a flow injection analyzer (Westco SmartChem 200 Discrete Analyzer), with a detection limit of 0.05 mg/L ⁸².

Statistical Analyses

Means and standard errors were calculated for aqueous and solid-phase chemical measurements in all cores using replicates described below. Half the detection limit was used when measured concentrations were below detection limits. Total elemental concentrations were measured in 3-4 solid-phase aliquots from each soil core (**Table S1**). At each soil depth interval (1-cm from 0-6 cm; 2-cm from 6-16 cm), triplicate aqueous extractions were conducted to evaluate reactive Cr(VI) and exchangeable NH_4^+ concentrations (**Figure 2, Figure 3a, Figure S10**). In select soil depths within cores, replicates were limited (less than 3) due to solid mass or post-extraction aqueous volume.

To assess data normality, we applied the Shapiro-Wilk test and reported W statistics and p-values (**Table S7**). If data met normality assumptions at the 95% confidence interval (p-value = 0.05), we used two-sided parametric tests; otherwise, we utilized two-sided nonparametric tests. Likewise, we used the f-test to determine equal variance. Unpaired t

tests were used to compare mean reactive Cr(VI) concentrations at the 95% confidence interval in near surface soil (0-2 cm) of fire-affected and unburned sites based on geology. If one or both datasets were not normally distributed, such as in burned and unburned soils at control depths (10-16 cm), Mann-Whitney U test was used. Within a soil core, we compared mean reactive Cr(VI) concentrations in surface soil (0-2 cm) versus control depths (10-16 cm) using either paired *t* test or Wilcoxon signed rank test. Detailed information about and results for each statistical analysis is provided in Tables S2-S5 and Table S7 the Supplementary Information. All statistical analyses were executed using the *stats* package in R (v. 4.1.3)."

From comments made on the manuscript pdf

Introduction section needs a short paragraph at the beginning to discuss elements distribution issues in wildfire impacted areas worldwide. More papers related to this paragraph will be beneficial for the paper.

May I suggest, among others, the following articles, e.g.:

- 1) *Wildfire effects on soil quality. Application on a suburban area of West Attica (Greece). Geosciences Journal*, 25 (2), 243–253 (<https://doi.org/10.1007/s12303-020-0011-1>).
- 2) *Suburban areas in flames: Dispersion of potentially toxic elements from burned vegetation and buildings. Estimation of the associated ecological and human health risk. Environmental Research*, 183, 109153, <https://doi.org/10.1016/j.envres.2020.109153>.
- 3) *Elements' Content in Stream Sediment and Wildfire Ash of Suburban Areas in West Attica (Greece). Water* 2022, 14, 310. <https://doi.org/10.3390/w14030310>

Response: We thank the reviewer for the suggestion to add a paragraph discussing elemental concentrations of soil and ash after wildfires. We address past work that has quantified elemental concentrations in soils and ash in the first two paragraphs of the *Introduction*. We have added the second reference the reviewer suggested to our main text, in addition to a few recent studies on metals and their prevalence as a function of structural burning. References 1 and 3 address elemental concentrations within stream sediments of WUIs, and are ancillary to the introduction on metals within airborne particulate matter, and soil and ash, which are the focus of our study.

Changes: Based on both reviewers' recommendations, we have added the following references within the main text:

10. Xu, R. et al. Wildfires, Global Climate Change, and Human Health. N. Engl. J. Med. 383, 2173–2181 (2020).

26. Jahn, L. G. et al. Metallic and crustal elements in biomass-burning aerosol and ash: Prevalence, significance, and similarity to soil particles. ACS Earth Sp. Chem. 5, 136–148 (2021).

27. Ré, A. et al. Cytotoxic effects of wildfire ashes: In-vitro responses of skin cells. *Environ. Pollut.* **285**, 117279 (2021).
29. Boaggio, K. et al. Beyond Particulate Matter Mass: Heightened Levels of Lead and Other Pollutants Associated with Destructive Fire Events in California. *Environ. Sci. Technol.* **56**, 14272–14283 (2022).
30. Alshehri, T. et al. Wildland-urban interface fire ashes as a major source of incidental nanomaterials. *J. Hazard. Mater.* **443**, 130311 (2023).
32. Alexakis, D. E. Suburban areas in flames: Dispersion of potentially toxic elements from burned vegetation and buildings. Estimation of the associated ecological and human health risk. *Environ. Res.* **183**, 109153 (2020).
33. Alam, M. et al. Identification and quantification of Cr, Cu, and As incidental nanomaterials derived from CCA-treated wood in wildland-urban interface fire ashes. *J. Hazard. Mater.* **445**, 130608 (2023).
41. Yu, Y. & Ginoux, P. Enhanced dust emission following large wildfires due to vegetation disturbance. *Nat. Geosci.* 2022 1511 **15**, 878–884 (2022).
42. Shakesby, R. A. & Doerr, S. H. Wildfire as a hydrological and geomorphological agent. *Earth-Science Rev.* **74**, 269–307 (2006).
55. Rascio, I. et al. Evidence of hexavalent chromium formation and changes of Cr speciation after laboratory-simulated fires of composted tannery sludges long-term amended agricultural soils. *J. Hazard. Mater.* **436**, 129117 (2022).
80. James, B. R. & Bartlett, R. J. Behavior of Chromium in Soils: VII. Adsorption and Reduction of Hexavalent Forms¹. *J. Environ. Qual.* **12**, 177 (1983).
81. Rai, D., Eary, L. E. & Zachara, J. M. Environmental chemistry of chromium. *Sci. Total Environ.* **86**, 15–23 (1989).
82. McClain, C. N., Fendorf, S., Johnson, S. T., Menendez, A. & Maher, K. Lithologic and redox controls on hexavalent chromium in vadose zone sediments of California's Central Valley. *Geochim. Cosmochim. Acta* **265**, 478–494 (2019).

Figure 2: Change A, B, and C in Figure plot panels to a, b, and c, reflected in the figure caption

Response: We appreciate the reviewer's feedback and have made the salient changes.

Changes: We have revised the figure panels (see below) in addition to references to the figure throughout the main text.

Please avoid using "we". Apply this type of revision all over the manuscript.

Response: We appreciate the reviewer's concern for the use of "we" within the manuscript. According to the editor and style guidelines for *Nature Communications*, the use of "we" is allowed and in some places encouraged.

Changes: No changes were made.

Please avoid using "we".

Response: Please refer to previous response.

Changes: No changes were made.

Figure 3: Change A, B, and C in Figure plot panels to a, b, and c, reflected in the figure caption

Response: We appreciate the reviewer's feedback agree with the suggestions.

Changes: We have revised the figure panels (see below) in addition to references to the figure throughout the main text.

Figure 4: Change A and B in Figure plot panels to a and b reflected in the figure caption

Response: We appreciate the reviewer's feedback and agree with the suggestions.

Changes: We have revised the figure panels (see below) in addition to references to the figure throughout the main text. Please note that this figure was moved to the Supplementary Information (revised **Figure S8**), based on the recommendation of Reviewer 1.

Where are the sections "Results and Discussion" and "Conclusions"?

Response: We have followed the format for *Nature*, which often do not have specific sections denoted as "Results and Discussion" nor "Conclusions". Further, within the author guidelines for *Nature Communication* submissions: "*Nature Communications* is flexible with regard to the format of initial submissions. Within reason, style and length will not directly influence consideration of a manuscript. We also do not require a particular structure or format at first submission. If and when revisions are required, the editor will provide detailed formatting instructions at that time." Based on the editor's instructions for manuscript revisions, we have defined "Introduction" and "Results" sections in the manuscript according to the *Nature Communications* formatting instructions.

Changes: Based on the editor's instructions for manuscript revisions, we have defined "Introduction" and "Results" sections in the manuscript according to the *Nature Communications* formatting instructions.